# Disentangling Likes and Dislikes in Personalized Generative Explainable Recommendation

## Abstract

Recent research on explainable recommendation generally frames the task as a standard text generation problem, and evaluates models simply based on the textual similarity between the predicted and ground-truth explanations. However, this approach fails to consider one crucial aspect of the systems: whether their outputs accurately reflect the users' (post-purchase) sentiments, i.e., whether and why they would like and/or dislike the recommended items. To shed light on this issue, we introduce new datasets and evaluation methods that focus on the users' sentiments. Specifically, we construct the datasets by explicitly extracting users' positive and negative opinions from their post-purchase reviews using an LLM, and propose to evaluate systems based on whether the generated explanations 1) align well with the users' sentiments, and 2) accurately identify both positive and negative opinions of users on the target items. We benchmark several recent models on our datasets and demonstrate that achieving strong performance on existing metrics does not ensure that the generated explanations align well with the users' sentiments. Lastly, we find that existing models can provide more sentiment-aware explanations when the users' (predicted) ratings for the target items are directly fed into the models as input. We will release our code and datasets upon acceptance.

## CCS Concepts

• **Information systems → Personalization**.

## Keywords

Explainable recommendation, Recommender systems, Large language model, Transformer, Personalization, Sentiment analysis

**ACM Reference Format:**
Anonymous Author(s). 2024. Disentangling Likes and Dislikes in Personalized Generative Explainable Recommendation. In *Proceedings of Make sure to enter the correct conference title from your rights confirmation email (Conference acronym 'XX)*. ACM, New York, NY, USA, 14 pages. https://doi.org/XXXXXXX.XXXXXXX

## 1 Introduction

Recently, there has been a growing interest in developing explainable recommendation systems, which not only recommend items to target users, but also provide *explanations* as to why they would like the recommended items [4, 5, 28, 41, 42, 60, 61]. To achieve this goal, most previous studies automatically extract users' main opinions about items from their post-purchase reviews, which they treat as ground-truth explanations, and train a model that generates the extracted texts given users and items as input [6, 19–22, 39, 53]. However, a majority of existing datasets are constructed using rudimentary algorithms, and they often discard users' important opinions and sentiments [3]. For instance, in the example presented in the first row of Table 1, only a positive opinion is extracted from the original review, which also describes the negative aspects of the item. Training models on such noisy data will result in poor performance, motivating the need to create a more reliable dataset.

Additionally, another limitation of previous studies is that they perform evaluation largely based on the string matching or textual similarity (e.g., as measured BERTScore [56]) between the model's outputs and the sentences or *features* (keywords) extracted from the reviews. However, this approach cannot take into account whether the model accurately predicts the *sentiments* (positive or negative) of the original reviews. That is, a model can achieve good scores as long as it generates a lot of keywords even if they are mentioned with the wrong sentiment. We argue that considering sentiments is vital in evaluation, since users can mention mixed feelings about one item in the review and hence predicting keywords alone does not suffice to provide reliable and convincing explanations.

To address the aforementioned limitations, we introduce new datasets that focus on *whether and why users like and/or dislike the recommended items*. To construct such datasets, we utilize a large language model (LLM) to: (1) summarize a user review; and (2) extract a list of positive and negative opinions (features) separately from the summary, i.e., what the user likes and dislikes about an item. Table 1 shows two examples of the generated summaries and extracted features — we treat the summaries as ground-truth explanations, and use the features to perform fine-grained evaluation. Specifically, we propose to evaluate models from two perspectives: whether the model's output (1) aligns well with the user's sentiment; and (2) correctly identifies the positive and negative features.

We evaluate several recent models using our datasets and evaluation methods, and find that strong models in existing metrics such as BERTScore do not necessarily capture the users' sentiments very well. Additionally, we find that existing models can generate more sentiment-aware explanations when we use the users' (predicted) ratings for the target items as additional input of the models.

In summary, our contributions are as follows:

- We introduce new datasets for explainable recommendations that focus on the users' sentiments. Using an LLM, we construct reliable datasets that explicitly present the users' positive and negative opinions about items.
- Using our datasets, we propose to evaluate models based on whether they accurately reflect the users' sentiments. We show that existing evaluation metrics are limited in measuring the sentiment alignment.

**Table 1: Examples of user reviews and ground-truth explanations (extracted from the reviews) in an existing dataset and ours.**

| Original review data | Existing explanation data | OURS |
|---|---|---|
| text: I'm back, did I miss anything? Hewitt is in college and trying to get on with her life when her friend wins a trip for 4 to the Bohamas. There, ... killer from part I and his son. Gets off to a great start, *but falls into the rut of predictability with an overdone body count.* | explanation: gets off to a **great start**, feature: start, opinion: great | explanation: User dislikes **predictability** and **excessive body count**, but appreciates the initial **engaging start**., positive features: ["engaging start"], negative features: ["predictability", "excessive body count"] |
| text: would you like some serial for breakfast? *Great movie, really outrageous, really shocking and Kathleen Turner gives a 5 star performance... she is terrific... it is really hard not to like this movie,* unless u have a really bad sense of humor. *Absolutely perfect.... a really fun time...* | explanation: unless u have a really **bad** sense of **humor**, feature: humor, opinion: bad | explanation: User enjoys **outrageous humor** and **strong performances**, particularly praising **Kathleen Turner's role** in the movie., positive features: ["outrageous humor", "strong performances", "Kathleen Turner's role"], negative features: [] |

- We find that the users' predicted ratings about items help models to generate more sentiment-aware explanations.

## 2 Related Work

Previous work on explainable recommendation extracts ground-truth explanations from either item descriptions [10] or user reviews [13, 15, 19, 20, 23, 45, 54], and we use the latter source to build our datasets for personalized recommendation. However, one problem is that user reviews can contain a lot of irrelevant information to the items or users' preferences, and hence existing work aims to mine the users' main opinions from reviews in various ways. For instance, Li et al. [20] extract sentences or phrases that appear frequently in all reviews throughout a dataset, but this approach often results in retrieving very short phrases that are too general to serve as explanations, e.g., *great movie.* Li et al. [23] make use of "tips" (i.e., short-text user reviews) as explanations, but they often lack important information about items. The most widely used dataset in existing research [6, 21, 22, 39, 53] is the one constructed by Li et al. [19]. They first identify features (i.e. aspects of an item) and users' opinions about them using a sentiment analysis toolkit, and then generate ground-truth explanations by retrieving a sentence that contains at least one feature from each review. The second column in Table 1 (under "Existing explanation data") shows two examples of the generated explanations, features, and opinions mined from the original reviews shown in the first column. As can be seen, the extracted explanations do not accurately reflect the users' opinions; E.g., in the first instance, only the positive opinion is extracted from the review that represents mixed sentiments, and in the second instance, *bad* is extracted as the only opinion despite the very positive tone of the original review.

Concurrent to our work, recent studies use LLMs to construct more reliable datasets for explainable recommendations. Ma et al. [31] generate explanations by feeding user reviews to GPT-3.5 and asking why the user would enjoy the target item. This simple approach, however, could ignore negative opinions when a review contains mixed sentiments. Chen et al. [3] construct a dataset by prompting LLMs to extract two aspects from reviews: (1) purchase reasons (e.g., *Birthday gift for a teenage daughter who likes AI features*); and (2) post-purchase experience (e.g., *The daughter loves the AI photo editor and found it a useful tool*). Using this dataset, they propose the tasks of predicting each aspect given item and user information as input. Compared to this work, we focus on extracting positive and negative opinions separately from user reviews, and propose to assess the model's ability to generate explanations with the correct sentiment.

## 3 Our Datasets

### 3.1 Dataset Construction

We construct new datasets for explainable recommendation from existing user review datasets. Our datasets are built in two steps: *review summarization* and *positive/negative feature extraction*.

In the review summarization step, we extract users' main opinions from reviews (and use them as ground-truth explanations) by prompting an LLM to explain what the user likes or dislikes about the target item, using the prompt shown in Table 2. For the LLM, we use GPT-4o-mini [36]. To reduce the risk of hallucinations and keep the explanations concise, we restrict the model's output to 15 words or less, which roughly aligns with the average lengths of the explanations in existing datasets.[1]

In the feature extraction step, we further prompt GPT-4o-mini to extract users' positive and/or negative opinions about items (denoted as *features*) from the explanations generated in the previous step; Table 3 shows the prompt used in this step. This feature extraction task is known as *aspect-based sentiment analysis* [33, 38, 59] in natural language processing, and recent studies demonstrate that LLMs perform well on this task even in zero-shot or few-shot settings [14, 17, 58]. Table 1 shows two examples of the generated explanations and extracted features under the OURS column. Compared to the existing dataset shown next to OURS, our dataset summarizes the reviews more accurately and also extracts the features along with the associated sentiments (either positive or negative). This new format makes it possible to perform more fine-grained evaluation based on whether a model generates explanations with the correct sentiment, as we will explain in Section 4.

We construct our datasets from three existing user review datasets in different domains, namely Amazon [34], Yelp [55], and RateBeer [32]. Amazon contains user reviews for movies; Yelp for restaurants; and RateBeer for alcoholic drinks. We discard very

---

[1]See Table 14 in Appendix for the details of existing datasets.

**Table 2: The prompt used for the review summarization task, followed by an input and output example.**

prompt: System: You are a smart recommender system.
Assistant: rating: <rating>/<max_rating>, review: <review_text>
User: Please explain within <n> words based on the rating and review of what the user likes or dislikes about the item,

input: <rating>=5, <max_rating>=5, <review_text>="I'm back, did I miss anything? Hewitt is in college and trying to get on with her life when her friend wins a trip for 4 to the Bohamas. There, they and a bunch of innocent bystanders are killed one by one by the killer from part I and his son. Gets off to a great start, but falls into the rut of predictability with an overdone body count.", <n>=15

output: "User dislikes predictability and excessive body count, but appreciates the initial engaging start."

**Table 3: The prompt used for the positive/negative features extraction task, followed by an input and output example.**

prompt: System: You are a helpful assistant.
Assistant: text: <text>
User: Please extract the features that the user likes or dislikes about the item from the text. The features must be included in the original text. Return the result in JSON format with the following structure: 'likes': ['feature_1,' 'feature_2'], 'dislikes': ['feature_3', 'feature_4']. Do not include any other sentences.

input: <text>="User dislikes predictability and excessive body count, but appreciates the initial engaging start."

output: {likes: ["engaging start"], dislikes: ["predictability", "excessive body count"]}

**Table 4: Statistics of three datasets used in our experiments.**

|  | Amazon [34] | Yelp [55] | RateBeer [32] |
|---|---|---|---|
| #users | 7,445 | 11,780 | 2,743 |
| #items | 7,331 | 10,148 | 7,452 |
| #interactions | 438,604 | 504,184 | 512,370 |
| #positive features | 10,676 | 8,826 | 5,672 |
| #negative features | 10,999 | 9,252 | 3,284 |
| #records / user | 58.91 | 42.79 | 186.79 |
| #records / item | 59.82 | 49.68 | 68.75 |
| #words / explanation | 13.72 | 13.71 | 13.76 |
| max rating | 5 | 5 | 20 |

short reviews that contain less than 15 words. Following previous work [18, 19], we also exclude the users/items which interact with the other items/users less than 20 times in the entire dataset. Table 4 shows the statistics of our datasets generated from each source. We use the latest and second latest interactions of each user as test and validation data, respectively, and use the rest as training data.

## 3.2 Dataset Quality Evaluation

While LLMs generally perform well on summarization [1, 7, 49, 57] and feature extraction [14, 17, 58], there is always a risk of

**Table 5: The human evaluation results on the dataset quality.**

| Stage | Type | Amazon | Yelp | RateBeer |
|---|---|---|---|---|
| 1 | Factual | 0.95 | 1.00 | 0.96 |
|  | Context-p | 0.98 | 0.97 | 0.99 |
|  | Context-n | 0.99 | 0.99 | 0.96 |
| 2 | Factual-p | 1.00 | 1.00 | 1.00 |
|  | Factual-n | 0.99 | 1.00 | 0.99 |
|  | Complete-p | 0.99 | 1.00 | 1.00 |
|  | Complete-n | 1.00 | 1.00 | 1.00 |

**Table 6: The results of the dataset quality evaluation using GPT-4o.** The numbers outside parentheses denote the scores estimated by GPT-4o, whereas those in parentheses indicate the percentage of the instances for which GPT-4o and human annotators make the same judgements.

| Stage | Type | Amazon | Yelp | RateBeer |
|---|---|---|---|---|
| 1 | Factual | 0.990 (0.95) | 0.993 (0.98) | 0.997 (0.95) |
|  | Context-p | 0.996 (0.98) | 0.997 (0.96) | 0.997 (0.98) |
|  | Context-n | 0.962 (0.97) | 0.971 (0.95) | 0.965 (0.97) |
| 2 | Factual-p | 0.999 (1.00) | 0.999 (1.00) | 0.996 (1.00) |
|  | Factual-n | 0.998 (0.99) | 0.998 (1.00) | 0.998 (0.99) |
|  | Complete-p | 0.997 (0.99) | 0.997 (1.00) | 0.998 (1.00) |
|  | Complete-n | 0.998 (1.00) | 0.996 (1.00) | 0.998 (1.00) |

hallucinations [11, 16, 25, 46]. An ideal solution to this problem is to verify the datasets by hiring human annotators, which however comes with a considerable annotation cost. Therefore, inspired by Chen et al. [3], we verify the dataset quality by utilizing GPT-4o [35] as an automated evaluator. To ensure its reliability, we also ask human annotators to assess a small portion of the datasets and measure the agreement between the humans and GPT-4o.

We evaluate the LLM's outputs generated at the "review summarization" and "positive/negative feature extraction" steps, respectively. We verify the summarizations (which we use as the ground-truth explanations) based on the following metrics:

- Factual hallucination (denoted as *Factual*): the percentage of the instances that do not contain any information that is not described or implied in the original reviews.
- Contextual hallucination for positive/negative features (denoted as *context-p/n*): the percentage of the instances where the positive/negative features are mentioned with the correct (not the opposite) sentiment.

For instance, given the user review: *I was fascinated by the romantic scenes*, a summary should be labeled as factual hallucination if it says *the user enjoys the thriller aspects*; and as contextual hallucination if it says *the user hates the romantic scenes*.

Then, we also verify the extracted positive and negative features based on the following metrics:

- Factual hallucination for positive/negative features (denoted as *factual-p/n*): the percentage of the instances that do not include any positive/negative features that are not present in the explanations.

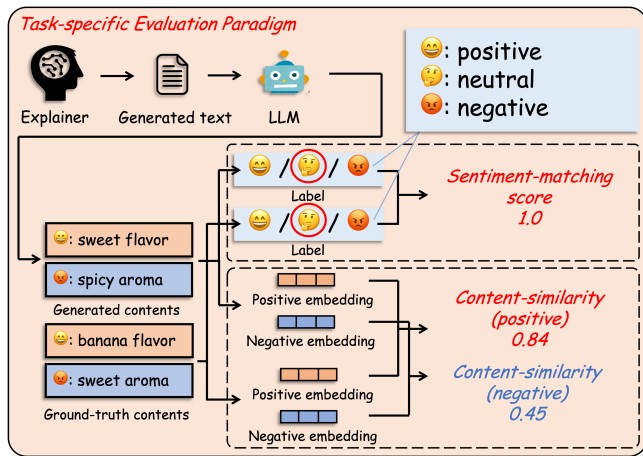

**Figure 1: An overview of how we calculate the sentiment-matching score and the content similarity of positive and negative features.**

- Completeness of positive/negative features (denoted as *complete-p/n*): the percentage of the instances that contain all positive/negative features mentioned in the explanations.

For instance, given the explanation: *the user enjoyed the thriller aspect and great action*, the model should flag factual hallucination if the extracted positive features contain *romantic* aspect; and a lack of completeness if they include *thriller aspect* only.

We sample 100,000 instances from each dataset generated in Section 3.1, and prompt GPT-4o to calculate the metrics described above (the exact prompts are provided in Tables 15 and 16 in Appendix). We also sample 100 instances among them for each dataset and ask five human annotators to perform the same evaluation (one annotator per review). We first present the results of the human evaluation in Table 5. The scores are very high across all metrics and datasets, indicating the high quality of our datasets. Next, Table 6 shows the results of the auto-evaluation using GPT-4o, where the numbers in brackets denote the percentage of the instances for which GPT-4o and the human annotators make the same judgements. The agreement scores are very high overall, verifying the effectiveness of GPT-4o as an automated evaluator. The table also shows that all datasets contain very few hallucinations, with the positive and negative features extracted correctly from the summaries. These results ensure the reliability and accuracy of our datasets.

## 4 Evaluation Methods

In previous work, models are evaluated based on standard textual similarity metrics, such as BLEU [37], ROUGE [26], and BERTScore [56]. Several studies [18, 19, 21, 22, 39] also look at whether the model's output contains a single-word feature included in the ground-truth explanation (e.g., *great* and *humor* in the existing data in Table 1). However, these evaluation metrics cannot consider whether the model predicts the correct *sentiments* of the original review. For example, if the ground-truth explanation is *the user loves the movie's storyline but is dissatisfied with the visual quality,*

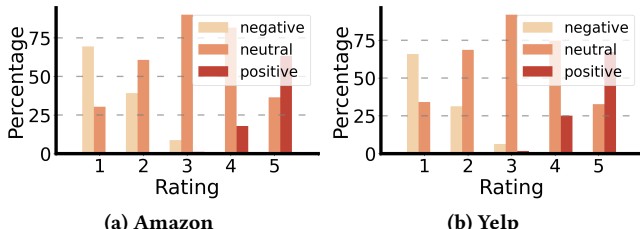

(a) Amazon      (b) Yelp

**Figure 2: Rating-sentiment distributions on the entire Amazon and Yelp datasets.**

and the generated explanation is *the user loves the visual quality but is dissatisfied with the movie's storyline*, previous metrics assign unreasonably high scores to the generated explanation due to the significant overlap of words and phrases between the two texts, including the key features *visual quality* and *movie's storyline*. However, the generated explanation does not accurately describe what the user would like and dislike about the movie, and providing such erroneous explanations for users will lead to losing their trust in the system.

To address this problem, we propose two evaluation metrics that focus on whether the generated explanations: (1) are consistent with the users' (post-purchase) sentiments; and (2) accurately identify the positive/negative features, respectively. We name the former measure as a **sentiment-matching score** (denoted as *sentiment*), and the latter as a **content similarity of the positive/negative features** (denoted as *content-p/n*). Figure 1 illustrates an overview of how we calculate these scores.

The sentiment-matching score measures the agreement of the sentiments between the generated and ground-truth explanations. We first input each explanation into GPT-4o-mini and extract both positive and negative features included in it. To this end, we use the same prompt as we used for the feature extraction step in Section 3.1, which we showed to be effective and accurate in Section 3.2. Next, we label the explanation as "0" if only negative features are extracted; "1" if both positive and negative features are extracted; and "2" if only positive features are extracted. Lastly, we measure the sentiment-matching score as the percentage of the instances for which the generated and ground-truth explanations have the same labels. Figure 2 illustrates the distributions of the sentiment labels assigned to the ground-truth explanations on Amazon and Yelp. On both datasets, the number of positive/negative labels increases/decreases as the users' ratings get higher, suggesting that GPT-4o-mini recognizes the sentiments very well.

The second metric *Content-p/n* measures the textual similarities of the positive/negative features between the generated and ground-truth explanations. As a similarity measure, we use BERTScore, which calculates the similarity between a pair of texts using a pre-trained language model.[2] When there are multiple positive (or negative) features, we concatenate them with *and* before calculating the similarity. Note that when both ground-truth and generated texts have no positive (or negative) features, we set Content-p (or Content-n) to 1.0, and when the ground-truth has positive/negative

---

[2]We use roberta-large [29] following the default configuration.

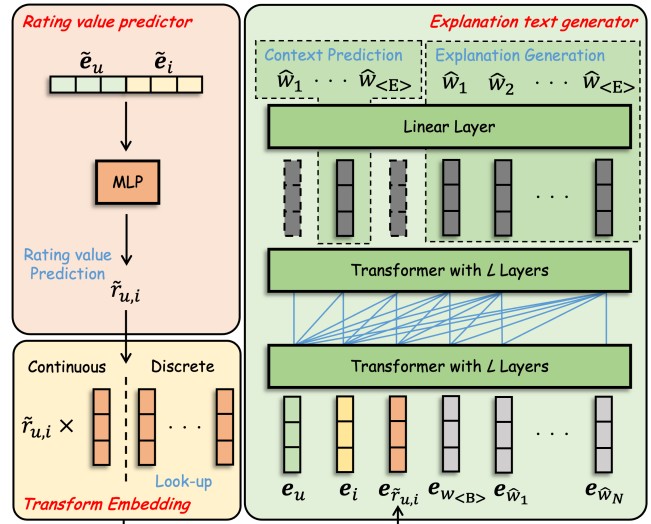

**Figure 3: An overview of PETER-c/d-emb.** Here, $u$ and $i$ denote the user and item indices, resp.; $\tilde{r}_{u,i}$ is the predicted rating of the $u$-th user for the $i$-th item; $\tilde{e}$ and $e$ denote separate input embeddings; $\hat{w}_j$ is the $j$-th predicted word; $N$ is the total number of the generated words; and /<E> denote the beginning/end of the sentence.

features but the generated one doesn't (and vice versa), we set the score to 0.0.

## 5 Evaluation Experiment

Using our proposed datasets, we benchmark recent models for explainable recommendation. We evaluate them using our proposed evaluation methods proposed in Section 4, as well as with several establish metrics such as BLEU and ROUGE.

### 5.1 Models

We evaluate various models listed in Table 7, which include CER [39], ERRA [6], PETER [21], and PEPLER/PEPLER-D [22]. All models are based on transformers [50] with or without pre-training on monolingual data, and are trained to generate explanations given user and item IDs as input.[3] Additionally, the models except for PEPLER-D also perform multi-task learning by predicting the users' ratings about the target items, which is found effective in enhancing the generation performance. Among these models, CER is trained with an auxiliary loss that minimizes the difference between the ratings predicted from the user and item IDs, and those from the hidden states of the explanation. The authors show that including this loss enhances the sentiment coherence between the predicted rating and explanation; e.g., the coherence is high if the model predicts a very high rating and generates a positive explanation such as *the movie is great.*

While the method used in CER is sensible, we hypothesize that directly feeding the predicted rating into the model as input would make it generate more coherent explanations with the rating, since this way the model can predict every word in the explanation conditioned directly on the rating information via self-attention. In

---

[3]The implementation details are in Appendix A.5.

**Table 7: Comparison of models used in our experiments.** "Output" means the model predicts users' ratings as a subtask, and "Input" means the model takes predicted ratings as input.

| Method | Pretrained | Rating | |
|--------|:----------:|:------:|:-----:|
| | | Output | Input |
| CER [39] | ✗ | ✓ | ✗ |
| ERRA [6] | ✗ | ✓ | ✗ |
| PETER [21] | ✗ | ✓ | ✗ |
| PEPLER [22] | ✓ | ✓ | ✗ |
| PEPLER-D [22] | ✓ | ✗ | ✗ |
| PETER-c/d-emb | ✗ | ✗ | ✓ |
| PEPLER-c/d-emb | ✓ | ✗ | ✓ |

fact, this approach was also adopted by earlier models [18, 24] based on Gated Recurrent Unit (GRU) [9]. To verify our hypothesis, we propose to slightly modify PETER and PEPLER and let them directly take the predicted ratings as input. Figure 3 shows an overview of the modified version of PETER.[4] We remove the multi-tasking component for rating prediction and instead input the embedding of the rating $e_{\tilde{r}_{u,i}}$ (with the rating $\tilde{r}_{u,i}$ predicted by a pre-trained external model) in addition to the user and item embeddings $e_u$ and $e_i$. The rating embedding $e_{\tilde{r}_{u,i}}$ is obtained in two ways: (1) multiplying $\tilde{r}_{u,i}$ by a trainable vector; or (2) rounding $\tilde{r}_{u,i}$ into the nearest integer and look up the corresponding trainable vector. We refer to the former approach as "(PETER/PEPLER)-**c-emb**" and the latter as "(PETER/PEPLER)-**d-emb**", respectively.[5]

To predict users' ratings, we train a simple multi-layer perceptron (MLP) model that predicts ratings given user and item IDs, following the network used for multi-tasking in PEPLER. Note that our rating prediction model is pre-trained independently from explainable recommendation models (i.e., PETER and PEPLER). Although the performance on rating prediction is not the main subject of this study, we expect that the higher the accuracy is, the better the explainable recommendation models would perform. Therefore, in our experiments, we also evaluate how much improvements we can get when we use the users' ground-truth ratings as input, which we report as "(PETER/PEPLER)-c/d-**emb+**".

### 5.2 Evaluation Metrics

We evaluate models using our evaluation metrics proposed in Section 4 (i.e., the sentiment-matching score and content similarity of positive/negative features). We also report the scores in several established metrics used in previous work [6, 18, 19, 21, 22, 39]. They are categorized into two groups, referred to as the *text quality* metric and *explainability* metric, respectively. The former evaluates the quality of the generated explanations, while the latter focuses on the quality of the predicted features in the explanations.

For the text quality metrics, we use **BLEU** [37], **ROUGE** [26], **Unique Sentence Ratio** (**USR**) [19], and **BERTScore** (**BERT**) [56]. BLEU and ROUGE measure the $n$-gram overlaps between the generated and ground-truth explanations, with BLEU focusing on precision and ROUGE on recall. We calculate BLEU with $n \in \{1, 4\}$

---

[4]PEPLER has the same structure except it doesn't have the context prediction part.
[5]Earlier works [18, 24] have taken the latter approach by converting decimal ratings into either two or six discrete values and training embeddings for each.

**Table 8: Results based on our proposed evaluation metrics.** The best scores among all models are **boldfaced**.

| Method | Amazon | | | Yelp | | | RateBeer | | |
|---|---|---|---|---|---|---|---|---|---|
| | sentiment ↑ | content-p ↑ | content-n ↑ | sentiment ↑ | content-p ↑ | content-n ↑ | sentiment ↑ | content-p ↑ | content-n ↑ |
| CER | 0.5364 | 0.7131 | 0.6122 | 0.5265 | 0.7603 | 0.5519 | 0.6266 | 0.7925 | 0.6592 |
| ERRA | 0.5327 | 0.7243 | 0.6005 | 0.5275 | 0.7665 | 0.5435 | 0.6481 | 0.8082 | 0.6611 |
| PEPLER-D | 0.2842 | 0.4576 | 0.4935 | 0.3111 | 0.5252 | 0.4559 | 0.4633 | 0.5760 | 0.5991 |
| PETER | 0.5445 | 0.7130 | 0.6198 | 0.5238 | 0.7620 | 0.5483 | 0.6277 | 0.7902 | 0.6592 |
| PETER-c-emb | 0.5636 | 0.7348 | 0.6145 | 0.5471 | 0.7675 | 0.5622 | 0.6591 | **0.8280** | 0.6540 |
| PETER-d-emb | 0.5695 | 0.7234 | 0.6251 | **0.5744** | **0.8099** | **0.5692** | 0.6445 | 0.8007 | 0.6629 |
| PEPLER | 0.5691 | 0.7532 | 0.6228 | 0.5462 | 0.8027 | 0.5470 | 0.6445 | 0.8061 | 0.6558 |
| PEPLER-c-emb | 0.5935 | 0.7682 | 0.6337 | 0.5624 | 0.8080 | 0.5562 | 0.6449 | 0.8075 | 0.6553 |
| PEPLER-d-emb | **0.5995** | **0.7717** | **0.6363** | 0.5539 | 0.8011 | 0.5536 | **0.6697** | 0.8163 | **0.6679** |

**Table 9: Results on Amazon based on evaluation metrics used in previous work.** The best scores among all models are **boldfaced**.

| Method | Text Quality | | | | | | Explainability | | | | | |
|---|---|---|---|---|---|---|---|---|---|---|---|---|
| | | | | | | | | Positive | | | Negative | |
| | B1 ↑ | B4 ↑ | R1 ↑ | R2 ↑ | USR ↑ | BERT ↑ | FMR ↑ | FCR ↑ | DIV ↓ | FMR ↑ | FCR ↑ | DIV ↓ |
| CER | 0.3729 | 0.0860 | 0.3934 | 0.1423 | 0.8964 | 0.8884 | 0.1886 | 0.2490 | 2.005 | 0.0939 | 0.2159 | 1.943 |
| ERRA | 0.3656 | 0.0793 | 0.3860 | 0.1354 | 0.5767 | 0.8892 | 0.1649 | 0.0960 | 2.568 | 0.0854 | 0.0876 | 2.420 |
| PEPLER-D | 0.0605 | 0.0024 | 0.1090 | 0.0086 | 0.6879 | 0.8434 | 0.0100 | 0.1816 | **0.128** | 0.0127 | 0.1785 | **0.088** |
| PETER | 0.3717 | 0.0859 | 0.3921 | 0.1422 | 0.8883 | 0.8883 | 0.1813 | 0.2387 | 1.994 | 0.0900 | 0.2078 | 1.919 |
| PETER-c-emb | 0.3730 | 0.0848 | 0.3927 | 0.1389 | 0.9176 | 0.8879 | **0.1918** | 0.2502 | 2.001 | 0.0911 | 0.2134 | 1.893 |
| PETER-d-emb | **0.3752** | **0.0867** | **0.3950** | **0.1424** | 0.9233 | 0.8895 | 0.1833 | 0.2457 | 2.055 | **0.0961** | 0.2167 | 1.980 |
| PEPLER | 0.3619 | 0.0797 | 0.3848 | 0.1350 | 0.9398 | 0.8882 | 0.1700 | 0.2495 | 2.066 | 0.0849 | **0.2320** | 1.995 |
| PEPLER-c-emb | 0.3689 | 0.0808 | 0.3882 | 0.1349 | 0.9474 | **0.8897** | 0.1718 | 0.2415 | 2.084 | 0.0848 | 0.2247 | 1.991 |
| PEPLER-d-emb | 0.3684 | 0.0777 | 0.3884 | 0.1337 | **0.9566** | 0.8894 | 0.1727 | **0.2510** | 2.125 | 0.0887 | 0.2308 | 2.072 |

(**B1** and **B4**) and ROUGE with $n \in \{1, 2\}$ (**R1** and **R2**), following previous work [6, 21, 22]. USR calculates the number of unique sentences generated by the model, divided by the total number of the generated sentences; the higher this score is, the more diverse the explanations are.

For the explainability metrics, we use **Feature Matching Ratio (FMR)**, **Feature Coverage Ratio (FCR)**, and **Feature Diversity (DIV)**. FMR measures the percentage of the explanations that include the ground-truth feature; and FCR and DIV measure the diversity of the generated features across all instances. These metrics are proposed by Li et al. [19] for evaluation on previous datasets where each ground-truth explanation contains only one *single-word* feature. To meet this requirement, in our experiments we randomly select one word from positive and/or negative features (which can contain a list of words or phrases) for each instance, and calculate the scores for each sentiment separately.[6]

## 6 Results and Analysis
### 6.1 Quantitative Results
Table 8 shows the results for each dataset based on our proposed evaluation metrics. It demonstrates that the models with our proposed modification (i.e., *-c/d-emb) outperform the original models and achieve the best scores on all datasets. These results verify our hypothesis that incorporating the users' predicted ratings as input is more effective than predicting the ratings as a subtask. We also find that the models that treat the ratings as discrete variables

---

[6]The details of each metric are shown in Appendix A.4.

**Table 10: Results on rating prediction.** The best scores among all models are **boldfaced**.

| Method | Amazon | | Yelp | | RateBeer | |
|---|---|---|---|---|---|---|
| | MAE↓ | RMSE↓ | MAE↓ | RMSE↓ | MAE↓ | RMSE↓ |
| PETER | **0.76** | 1.07 | **0.77** | 1.04 | 1.48 | 2.07 |
| PEPLER | 0.79 | **1.06** | 0.80 | 1.05 | 1.50 | 2.06 |
| PETER-c/d-emb PEPLER-c/d-emb | 0.78 | **1.06** | **0.77** | 1.03 | 1.46 | **2.01** |

(i.e., *-d-emb) generally perform better than those that treat them as continuous ones (i.e., *-c-emb). This is likely because there is a non-linear relationship between the users' sentiments and ratings about items, as we showed in Figure 2 in Section 4. When we look at the results on each dataset, PEPLER-d-emb achieves the best scores on Amazon and RateBeer but underperforms PETER-d-emb on Yelp. This demonstrates that the effectiveness of pre-training varies depending on the dataset (note that PEPLER fine-tunes GPT-2 but PETER is trained from scratch).

Table 9 presents the results on Amazon in existing metrics; we observe similar trends on Yelp and RateBeer and hence present the results in Table 17 and 18 in Appendix due to the limited space. The table shows that while PETER-d-emb performs the best on the text quality metrics, the improvements from PETER are marginal. Besides, on the explainability metrics, our modification does not enhance the performance of the original models very much. These results suggest that the existing metrics cannot properly evaluate

**Table 11: The performance gains/losses in our evaluation metrics when we use the ground-truth ratings as input (shown with "+").** The best scores of all models are underlined, and the gains/losses are marked in ↑**green** and ↓**red**, respectively.

| Method | Amazon | | | Yelp | | | RateBeer | | |
|---|---|---|---|---|---|---|---|---|---|
| | sentiment ↑ | content-p ↑ | content-n ↑ | sentiment ↑ | content-p ↑ | content-n ↑ | sentiment ↑ | content-p ↑ | content-n ↑ |
| PETER-d-emb | 0.5695 | 0.7234 | 0.6251 | 0.5744 | 0.8099 | 0.5692 | 0.6445 | 0.8007 | 0.6629 |
| PETER-d-emb+ | _0.6259_ ↑9.9% | _0.7575_ ↑4.7% | _0.6816_ ↑9.0% | _0.6609_ ↑15.0% | _0.8304_ ↑2.5% | _0.6514_ ↑14.4% | 0.6616 ↑2.6% | 0.7970 ↓0.4% | _0.6971_ ↑5.1% |
| PEPLER-d-emb | 0.5995 | 0.7717 | 0.6363 | 0.5539 | 0.8011 | 0.5536 | _0.6697_ | _0.8163_ | 0.6679 |
| PEPLER-d-emb+ | _0.6503_ ↑8.4% | _0.7790_ ↑0.9% | _0.7006_ ↑10.1% | 0.6488 ↑17.1% | 0.7783 ↓2.8% | 0.6267 ↑13.2% | 0.6653 ↓0.6% | 0.8044 ↓1.4% | 0.6876 ↑2.9% |

**Table 12: The performance gains/losses in existing metrics on Amazon when we use the ground-truth ratings as input (shown with "+").** The best scores of all models are underlined, and the gains/losses are marked in ↑**green** and ↓**red**, respectively.

| Method | Text Quality | | | | | | Explainability | | | | | |
|---|---|---|---|---|---|---|---|---|---|---|---|---|
| | | | | | | | | Positive | | | Negative | |
| | B1 ↑ | B4 ↑ | R1 ↑ | R2 ↑ | USR ↑ | BERT ↑ | FMR ↑ | FCR ↑ | DIV ↓ | FMR ↑ | FCR ↑ | DIV ↓ |
| PETER-d-emb | 0.3752 | 0.0867 | 0.3950 | 0.1424 | 0.9233 | 0.8895 | 0.1833 | 0.2457 | 2.055 | 0.0961 | 0.2167 | 1.980 |
| PETER-d-emb+ | _0.3906_ | _0.0988_ | _0.4111_ | _0.1641_ | 0.9255 | _0.8916_ | _0.1946_ | 0.2606 | 1.982 | _0.1029_ | 0.2285 | _1.924_ |
| | ↑4.1% | ↑13.9% | ↑15.2% | ↑0.2% | ↑0.2% | ↑0.2% | ↑6.1% | ↑6.0% | ↑3.5% | ↑7.0% | ↑5.4% | ↑2.8% |
| PEPLER-d-emb | 0.3684 | 0.0777 | 0.3884 | 0.1337 | 0.9566 | 0.8894 | 0.1727 | 0.2510 | 2.125 | 0.0887 | 0.2308 | 2.072 |
| PEPLER-d-emb+ | 0.3762 | 0.0912 | 0.3985 | 0.1545 | _0.9668_ | 0.8903 | 0.1687 | _0.2782_ | _1.958_ | 0.0918 | _0.2585_ | 1.989 |
| | ↑2.1% | ↑17.3% | ↑2.6% | ↑15.5% | ↑1.0% | ↑0.0% | ↓2.3% | ↑10.8% | ↑7.8% | ↑3.4% | ↑12.0% | ↑4.0% |

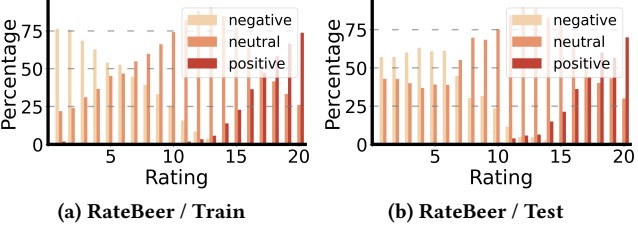

**(a) RateBeer / Train**     **(b) RateBeer / Test**

**Figure 4: Rating-sentiment distributions on the train and test sets of RateBeer.**

the alignment of the sentiments between the generated and ground-truth explanations; this does not come as a surprise given that the scores are based on the naive string matching or textual similarity.[7] On the other hand, our evaluation methods (and datasets) explicitly focus on users' sentiments, and we argue that reflecting them in the explanations is crucial to build reliable recommendation systems. Lastly, another interesting observation from Table 9 is that PETER performs better than ERRA and PEPLER overall, and that contradicts the previous findings that the latter models perform better on previous datasets [6, 22]. This suggests that optimal models differ depending on the nature of the dataset.

## 6.2 Performance on Rating Prediction

As we mentioned in Section 5.1, we propose to pre-train a rating prediction model and use its predictions as additional input of PETER and PEPLER. On the other hand, the original models of PETER and PEPLER predict ratings as a subtask. Intuitively, training a

---

[7]In particular, BERTScore assigns high scores to all models likely because our ground-truth explanations follow a similar format (e.g., *user likes ... but dislikes* ...), and the models can easily predict the high-frequency words; see Table 13 for some examples.

model specifically for rating prediction would lead to better performance on this task, and that could be part of the reasons why our proposed method works well. To investigate this, we compare the rating prediction performance among these models, and the results are presented in Table 10. We compare the performance in two metrics: **mean absolute error (MAE)** and **root mean square error (RMSE)**, both of which measure the distance between the predicted and ground-truth ratings. The table shows that in fact all models perform very similarly, demonstrating that our method benefits from using the ratings as input, rather than from training a separate model for rating prediction.

Next, we also analyse how much improvements we can get when we use the ground-truth ratings as input instead of the predicted ones, and Table 11 shows the results in our proposed metrics (the models that use the ground-truth data are shown with "+"). We can see that using the ground-truth ratings substantially improves performance on Amazon and Yelp for both PETER and PEPLER, indicating that the accuracy of rating prediction has a significant impact on generation performance. In contrast, we observe small or no improvements on RateBeer, and we attribute this to the fact that there is a discrepancy in the sentiment distributions between the train and test sets. Figure 4 compares the distributions of the sentiment labels assigned by GPT-4o-mini during our evaluation process described in Section 4. It shows that, on the training data, the percentage of negative labels decreases as the rating increases from 1 to 6, whereas the number remains nearly the same on the test set. On Amazon and Yelp, in contrast, we observe consistent patterns between the training and test sets, which we show in Figure 7 in Appendix.

Additionally, we also report the scores in the existing metrics on Amazon with or without the ground-truth ratings in Table 12. It demonstrates that using the ground-truth ratings as input also improves performance on all established metrics except FMR for

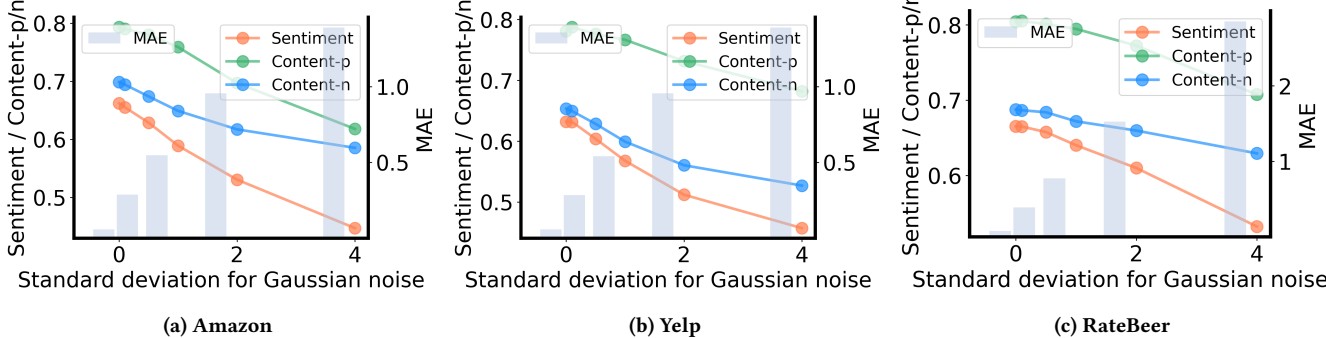

Figure 5: Simulation results on how the performance of PEPLER-d-emb+ changes when the ground-truth ratings used as input are distorted by Gaussian noise with different standard deviations.

Table 13: Two examples of the ground-truth and generated explanations on Amazon and Yelp. The words included in the ground-truth positive and negative features are colored in **red** and **blue**, respectively.

| Ground-truth (Amazon) | user enjoys **action** and **character development** but finds the **plot convoluted** with **multiple villains**. |
|---|---|
| CER | user dislikes the films lack of depth and **character development** despite appreciating garfields performance. |
| ERRA | user appreciates the engaging story and strong performances but dislikes the lack of **action**. |
| PETER | user dislikes the weak **villain** and **character development** but appreciates garfields portrayal. |
| PETER-d-emb | user appreciates **character development** and emotional depth but dislikes the **villains** portrayal and **villains**. |
| PEPLER | user dislikes the weak plot and **character development** despite appreciating the casts performances. |
| PEPLER-d-emb | user dislikes the **villains** and pacing but appreciates the romance and **character development**. |
| Ground-truth (Yelp) | user liked the **seafood quality** and **service**; disliked the **wait time** and **pricing**. |
| CER | user loves the delicious food friendly **service** and vibrant atmosphere no dislikes mentioned. |
| ERRA | user loves the food **quality** and **service** but dislikes the loud atmosphere and noise level. |
| PETER | user loves the delicious food friendly staff and vibrant atmosphere no dislikes mentioned. |
| PETER-d-emb | user loves the delicious food and friendly **service** but dislikes the long **wait time**. |
| PEPLER | user loves the delicious food friendly **service** and vibrant atmosphere dislikes nothing mentioned. |
| PEPLER-d-emb | user loves the delicious food and drinks but dislikes the long **wait** for **service**. |

PEPLER-d-emb, highlighting the relevance of the rating prediction task to explainable recommendation models.

Lastly, to further analyse the influence of the rating prediction accuracy, we add a Gaussian noise to the ground-truth ratings with different standard deviations and see how it affects the performance of PEPLER-d-emb+. Figure 5 shows the results, illustrating that the performance degrades sharply as the noise gets larger, especially on Amazon and Yelp. On the other hand, the impact is smaller on RateBeer, which is again likely due to the differences of the sentiment distributions between the train and test sets.

## 6.3 Case Studies

In Table 13, we present two examples of the ground-truth and generated explanations by CER, ERRA, PETER, PETER-d-emb, PEPLER and PEPLER-d-emb, respectively. In the first instance, PETER correctly identifies two features *character development* and *villains*, but wrongly predicts them both as negative features despite *character development* being mentioned positively in the ground-truth explanation. On the other hand, both PETER-d-emb and PEPLER-d-emb successfully generate these features with the correct sentiments. In the second example, only PETER-d-emb identifies the positive and

negative features (*service* and *wait time*, resp.) with the correct sentiments. These examples highlight the importance of considering the users' sentiments when we evaluate the quality of explanations. Our proposed datasets and metrics shed light on this problem, and open up a new research direction for explainable recommendation systems.

## 7 Conclusion

This paper introduced new datasets for explainable recommendations that focus on the users' sentiments. Using an LLM, we built reliable datasets in a new format that separately presents the users' positive and negative opinions about items. Based on our datasets, we introduced evaluation methods that focus on how well a model captures the users' sentiments. We benchmark various models on our datasets and find that existing evaluation metrics are limited in measuring the sentiment alignment between the generated and ground-truth explanations. Lastly, we found that we can make existing models more sensitive to the sentiments by feeding the users' predicted ratings about the target items as additional input of the models, and also showed that the rating prediction accuracy has a large impact on the quality of the generated explanations.

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

**Table 14: Statistics of the three existing datasets used in previous work [20].**

|  | **Amazon** | **Yelp** | **Tripadvisor [48]** |
|---|---|---|---|
| #users | 7,506 | 27,147 | 9,765 |
| #items | 7,360 | 20,266 | 6,280 |
| #interactions | 441,783 | 1,293,247 | 320,023 |
| #features | 5,399 | 7,340 | 5,069 |
| #records / user | 58.86 | 47.64 | 32.77 |
| #records / item | 60.02 | 63.81 | 50.96 |
| #words / explanation | **14.14** | **12.32** | **13.01** |
| max rating | 5 | 5 | 5 |

## A  Appendix

### A.1  Statistics of Existing Datasets

Table 14 shows the statistics of existing datasets [19] that are widely used in previous work on explainable recommendation [6, 21, 22, 39, 53]. Based on the average lengths of the explanations on these datasets, we restricted the output length of GPT-4o-mini to 15 or less words when summarizing user reviews.

### A.2  Dataset Quality Evaluation

Tables 15–16 show the prompts with input and output examples used for the dataset quality evaluation in Section 3.2. We use GPT-4o as an auto-evaluator of the outputs of GPT-4o-mini at the *review summarization* and *positive/negative feature extraction* steps, respectively. We design these prompts and the evaluation processes based on the methods proposed by Chen et al. [3].

### A.3  Generated Data Analysis

Figure 6 shows the users' rating distributions on Amazon, Yelp, and RateBeer. On Amazon and Yelp, users tend to assign high scores, while on RateBeer a majority of users give ratings between 10 and 20 and the distribution peaks at 15.

Figure 7 shows the rating-sentiment distributions on the train and test datasets of Amazon and Yelp. The distributions are similar between the train and test sets on these datasets, unlike on RateBeer (as we showed in Figure 4 in Section 6.2).

### A.4  Details of Existing Evaluation Metrics

USR calculates the number of the unique sentences generated by a model, divided by the total number of the sentences, as follows:

$$USR = \frac{|\mathcal{E}|}{N_{D_t}},\qquad(1)$$

where $\mathcal{E}$ denotes the set of unique sentences generated by a model, and $N_{D_t}$ is the total number of the instances on test data.

FMR calculates the percentage of the explanations that include the ground-truth feature, as follows:

$$FMR = \frac{1}{N_{D_t}} \sum_{u,i} \delta(f_{u,i} \in \hat{E}_{u,i}),\qquad(2)$$

where $f_{u,i}$ denotes the ground-truth feature; $\hat{E}_{u,i}$ denotes the generated explanation for the pair of the user $u$ and item $i$; and $\delta(x)$ is an indicator function which returns 1 if $x$ is true and 0 otherwise.

**Table 15: The prompt used for the auto-evaluation of the review summarization process, followed an input and output example.**

prompt: As a customer engagement team leader at Amazon, your task involves evaluating a summary written by a specialist about why a certain purchase was made. You will analyze the summary based on the provided customer review and rating, using these criteria:

1. hallucination: Answer "Hallucination" if the summary includes any unrelated features not mentioned by the customer review; otherwise, "Factual".
2. hallucination_reason: Provide a concise explanation for your assessment of the summary's hallucination.
3. context_positive: Answer "Hallucination" or "Correct". "Hallucination" if the summary includes any feature mentioned as a negative feature in the customer review as positive; otherwise, "Correct".
4. context_positive_reason: Provide a concise explanation for your assessment of the summary's hallucination.
5. context_negative: Answer "Hallucination" or "Correct". "Hallucination" if the summary includes any feature mentioned as a positive feature in the customer review as negative; otherwise, "Correct".
6. context_negative_reason: Provide a concise explanation for your assessment of the summary's hallucination.

Please respond using a valid json format, for example: {
"hallucination": "Factual",
"hallucination_reason": "...",
"context_positive": "Correct",
"context_positive_reason": "...",
"context_negative": "Correct",
"context_negative_reason": "...",
}

Now, please evaluate the following summary based on the above criteria:
Customer review: <review_text>.
Rating: <rating> / <max_rating>.
Specialist's summary of the review: <explanation_text>.

Assessment:

input: <review_text>="A must if you're in Nashville! Hot chicken is iconic to the city. The food is tasty, quick, and relatively cheap (I spent about $10 and felt full). The parking is not that great, but overall the restaurant itself is great.", <rating>=5, <max_rating>=5, <explanation_text>="User loves the tasty hot chicken, quick service, and affordability; dislikes limited parking.".

output: {
"hallucination": "Factual",
"hallucination_reason": "The summary accurately reflects the features mentioned in the customer review without adding unrelated features.",
"context_positive": "Correct",
"context_positive_reason": "The summary correctly identifies the positive aspects of the review, such as tasty food, quick service, and affordability.",
"context_negative": "Correct",
"context_negative_reason": "The summary correctly identifies the negative aspect of the review, which is the limited parking.",
}

FCR and DIV measure the diversity of the generated features across all instances. FCR is calculated as follows:

$$FCR = \frac{|\mathcal{F}_g|}{|\mathcal{F}|}, \tag{3}$$

where $\mathcal{F}$ is the set of unique features in the ground-truth explanations, and $\mathcal{F}_g$ denotes the set of the unique features included across all the generated explanations.

DIV calculates the diversity of features between the generated explanations. Specifically, this metric calculates the intersection of features between any pairs of two generated explanations, as follows:

$$DIV = \frac{2}{N_{D_t}(N_{D_t} - 1)} \sum_{u,u',i,i'} |\hat{\mathcal{F}}_{u,i} \cap \hat{\mathcal{F}}_{u',i'}|, \tag{4}$$

where $\hat{\mathcal{F}}_{u,i}$ denotes the feature set included in the generated explanation for the pair of the user $u$ and item $i$, and $\hat{\mathcal{F}}_{u',i'}$ for the pair of the user $u'$ and item $i'$, respectively.

## A.5 Implementation Details

In PETER, CER, and ERRA, we employ Stochastic Gradient Descent (SGD) [40] as the optimizer, with a batch size of 128 and an initial

**Table 16: The prompt used for the auto-evaluation of the feature extraction process, followed an input and output example.**

prompt: As a customer engagement team leader at Amazon, your task involves evaluating the positive and negative feature lists extracted from the explanation text about a user's experience after purchasing a product. You will check the positive and negative feature lists based on the provided explanation text, using these criteria:

1. hallucination_positive: Answer "Hallucination" if the positive feature list includes any unrelated features not mentioned by the explanation text; otherwise, "Factual".
2. hallucination_positive_reason: Provide a concise explanation for your assessment of the hallucination in the positive feature list.
3. completness_positive: "Yes" or "No". "Yes" if the positive feature list successfully includes all the positive features mentioned in the explanation text; otherwise, "No".
4. completness_positive_reason: Provide a concise explanation for your assessment of the positive feature list's completeness.
5. hallucination_negative: Answer "Hallucination" if the negative feature list includes any unrelated features not mentioned by the explanation text; otherwise, "Factual".
6. hallucination_negative_reason: Provide a concise explanation for your assessment of the hallucination in the negative feature list.
7. completness_negative: "Yes" or "No". "Yes" if the negative feature list successfully includes all the negative features mentioned in the explanation text; otherwise, "No".
8. completness_negative_reason: Provide a concise explanation for your assessment of the negative feature list's completeness.

Please respond using a valid json format, for example: {
"hallucination_positive": "Factual",
"hallucination_positive_reason": "...",
"completness_positive": "Yes",
"completness_positive_reason": "...",
"hallucination_negative": "Factual",
"hallucination_negative_reason": "...",
"completness_negative": "Yes",
"completness_negative_reason": "...",
}

Now, please evaluate the following positive and negative feature lists based on the above criteria:
Positive feature list: <features_positive>.
Negative feature list: <features_negative>.
Explanation text: <explanation_text>.

Assessment:

input: <features_positive>=["tasty hot chicken", "quick service", "affordability"], <features_negative>=["limited parking"], <explanation_text>="User dislikes predictability and excessive body count, but appreciates the initial engaging start."

output: {
"hallucination_positive": "Factual",
"hallucination_positive_reason": "All positive features listed ('tasty hot chicken', 'quick service', 'affordability') are mentioned in the explanation text.",
"completness_positive": "Yes",
"completness_positive_reason": "The positive feature list includes all the positive features mentioned in the explanation text.",
"hallucination_negative": "Factual",
"hallucination_negative_reason": "The negative feature 'limited parking' is mentioned in the explanation text.",
"completness_negative": "Yes",
completness_negative_reason": "The negative feature list includes all the negative features mentioned in the explanation text.",
}

learning rate of 1.0. During the training process, the learning rate is reduced by a factor of 0.25 if the validation loss does not improve, and the gradient clipping is applied with a maximum norm of 1.0 to stabilize the training process. The model architecture includes a multi-head attention (MHA) mechanism with two attention heads, each with 2048 units, and a dropout rate of 0.2 to prevent overfitting. In PETER and CER, we set the dimensionality of the embedding representation to 512; the number of MHA layers to 2; the weights for explanation generation regularization $\lambda_e$ and context regularization $\lambda_c$ to 1.0; and the rating regularization $\lambda_r$ to 0.1. In ERRA,

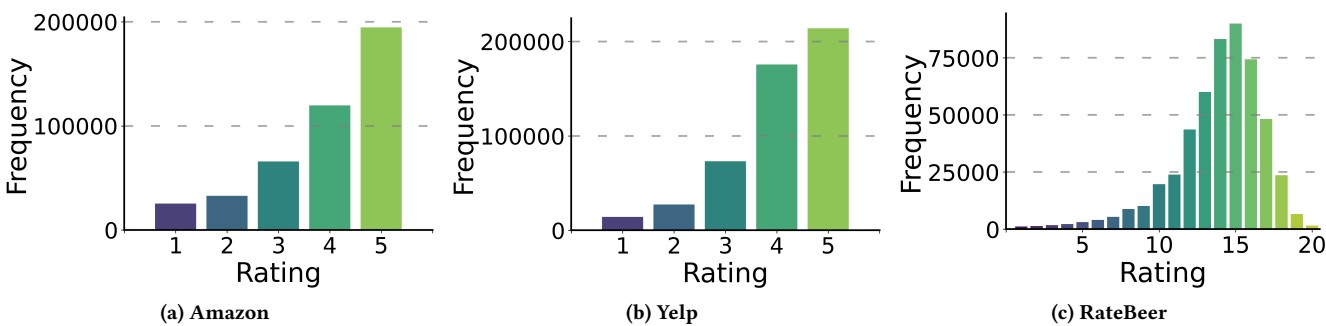

Figure 6: Rating distribution.

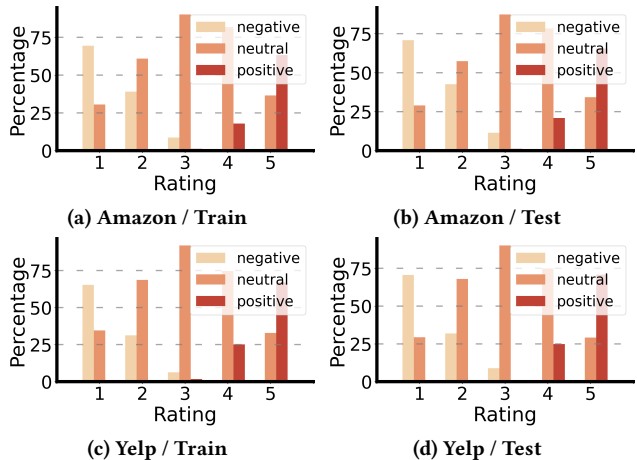

(a) Amazon / Train

(b) Amazon / Test

(c) Yelp / Train

(d) Yelp / Test

Figure 7: Rating-sentiment distribution on the train and test sets of Amazon and Yelp.

we set the dimensionality of the embedding representation to 384 and the number of MHA layers set to 6. We set the weight for the explanation regularization $\lambda_e$ to 1.0, context regularization $\lambda_c$ to 0.8, and the rating regularization $\lambda_r$ to 0.2. For CER, we exclude the ground-truth features of the target items from the model's input, as we do not assume that they are available during inference on the test set in our experiments.

In PEPLER and PEPLER-D, we use the pre-trained GPT-2 model as the foundation model of our architecture. The explanation generation regularization term $\lambda_e$ is set to 1.0 during training. During the training process, Adam [12] with decoupled weight decay [30] is used, with a batch size of 128, and the training process is stopped if the validation loss does not improve for five consecutive epochs. The optimizer uses a learning rate of 0.001/0.0001 for PEPLER/PEPLER-D and a weight decay of 0.01. In PEPLER, for the rating prediction network, we employ a multi-layer perceptron (MLP) with two hidden layers, each consisting of 400 units, and the rating regularization $\lambda_r$ is set to 0.01. For PEPLER-D, the number of retrieved feature words (which are used as the model's input) is set to 3.

For PETER-c/d-emb and PEPLER-c/d-emb, the experimental setups for training the generation models are the same as the ones used for PETER and PEPLER, respectively. To train a rating prediction model used by PETER-c/d-emb and PEPLER-c/d-emb, we employ SGD as the optimizer. The training is conducted with a batch size of 512, a learning rate of $1 \times 10^{-5}$, and a weight decay of 0.01. The model architecture consists of a two-layer MLP, each containing 400 units, with the dimensionality of the embedding representation set to 512.

## A.6 Results on Yelp and RateBeer in Existing Metrics

Tables 17–18 show the results in the standard metrics on Yelp and RateBeer datasets. These tables show that our modification does not lead to better performance in those metrics. These results suggest that the existing metrics cannot properly evaluate the alignment of the sentiments between the generated and ground-truth explanations.

## A.7 Recommendation Performance

Tables 19–20 show the results in the existing metrics on Yelp and RateBeer with or without using the ground-truth ratings. The tables demonstrate that using the ground-truth ratings as input improves performance on both datasets.

## A.8 Future Work

Future endeavors would involve improving the accuracy of rating prediction using more advanced models or additional information (e.g., item descriptions in text), as we showed that it has a large impact on the performance in both our proposed and existing evaluation metrics. It would also be intriguing to explore the application of LLMs to explainable recommendation systems in zero-shot or few-shot setups, as done by recent work [3, 27, 28, 52]. Another direction is to improve performance when the distributions are somewhat different between the train and test sets.

Following the trend of using LLMs for automated evaluation [2, 8, 43, 44, 47, 51] and inspired by the methods proposed by Chen et al. [3], we used GPT-4o to validate the quality of our datasets. However, since our methods could not detect all hallucinations included in our datasets, improving this process is a key to creating more reliable datasets.

Received 20 February 2007; revised 12 March 2009; accepted 5 June 2009

**Table 17: Results on Yelp based on evaluation metrics used in previous work.** The best scores among all models are **boldfaced**.

| Method | Text Quality | | | | | | Explainability | | | | | |
| | | | | | | | | Positive | | | Negative | |
| | B1 ↑ | B4 ↑ | R1 ↑ | R2 ↑ | USR ↑ | BERT ↑ | FMR ↑ | FCR ↑ | DIV ↓ | FMR ↑ | FCR ↑ | DIV ↓ |
|---|---|---|---|---|---|---|---|---|---|---|---|---|
| CER | 0.3879 | 0.0781 | 0.4046 | 0.1324 | 0.7343 | 0.8882 | 0.2312 | 0.2194 | 2.421 | 0.1229 | 0.2002 | 1.393 |
| ERRA | **0.3951** | 0.0780 | **0.4105** | **0.1327** | 0.3187 | **0.8897** | **0.2374** | 0.0900 | 3.169 | **0.1272** | 0.0851 | 2.141 |
| PEPLER-D | 0.0646 | 0.0013 | 0.1065 | 0.0040 | 0.6920 | 0.8370 | 0.0161 | **0.3088** | **0.168** | 0.0128 | **0.3182** | **0.163** |
| PETER | 0.3884 | **0.0793** | 0.4045 | 0.1326 | 0.7489 | 0.8882 | 0.2343 | 0.2216 | 2.441 | 0.1202 | 0.2020 | 1.372 |
| PETER-c-emb | 0.3895 | 0.0762 | 0.4074 | 0.1310 | 0.7383 | 0.8889 | 0.2332 | 0.2117 | 2.330 | 0.1212 | 0.1923 | 1.354 |
| PETER-d-emb | 0.3861 | 0.0716 | 0.4049 | 0.1268 | 0.7187 | 0.8882 | 0.2300 | 0.1991 | 2.232 | 0.1223 | 0.1813 | 1.372 |
| PEPLER | 0.3829 | 0.0732 | 0.3998 | 0.1277 | 0.8512 | 0.8875 | 0.2315 | 0.2517 | 2.425 | 0.1182 | 0.2360 | 1.372 |
| PEPLER-c-emb | 0.3769 | 0.0656 | 0.3957 | 0.1192 | 0.8801 | 0.8876 | 0.2080 | 0.2577 | 2.142 | 0.1172 | 0.2514 | 1.286 |
| PEPLER-d-emb | 0.3768 | 0.0696 | 0.3934 | 0.1233 | **0.9277** | 0.8859 | 0.2199 | 0.2719 | 2.227 | 0.1081 | 0.2590 | 1.166 |

**Table 18: Results on RateBeer based on evaluation metrics used in previous work.** The best scores among all models are **boldfaced**.

| Method | Text Quality | | | | | | Explainability | | | | | |
| | | | | | | | | Positive | | | Negative | |
| | B1 ↑ | B4 ↑ | R1 ↑ | R2 ↑ | USR ↑ | BERT ↑ | FMR ↑ | FCR ↑ | DIV ↓ | FMR ↑ | FCR ↑ | DIV ↓ |
|---|---|---|---|---|---|---|---|---|---|---|---|---|
| CER | 0.4349 | 0.1201 | 0.4718 | 0.1876 | 0.7036 | 0.9071 | 0.2841 | 0.1124 | 1.585 | 0.1280 | 0.0696 | 1.271 |
| ERRA | 0.4332 | 0.1168 | 0.4689 | 0.1852 | 0.5300 | 0.9071 | 0.2739 | 0.0822 | 1.714 | 0.1315 | 0.0513 | 1.477 |
| PEPLER-D | 0.1338 | 0.0127 | 0.1902 | 0.0368 | 0.4167 | 0.8582 | 0.0584 | 0.1413 | **0.348** | 0.0509 | 0.0883 | **0.659** |
| PETER | 0.4338 | 0.1191 | 0.4701 | 0.1856 | 0.7200 | 0.9067 | 0.2826 | 0.1109 | 1.552 | 0.1260 | 0.0701 | 1.258 |
| PETER-c-emb | **0.4374** | 0.1209 | 0.4722 | **0.1892** | 0.8519 | 0.9070 | **0.2851** | 0.1435 | 1.576 | 0.1290 | 0.0904 | 1.150 |
| PETER-d-emb | 0.4364 | **0.1214** | **0.4728** | 0.1885 | 0.8239 | **0.9073** | 0.2778 | 0.1294 | 1.448 | 0.1322 | 0.0818 | 1.195 |
| PEPLER | 0.4353 | 0.1177 | 0.4709 | 0.1864 | 0.8512 | 0.9067 | 0.2762 | 0.1493 | 1.657 | **0.1410** | 0.0950 | 1.198 |
| PEPLER-c-emb | 0.4356 | 0.1134 | 0.4722 | 0.1842 | 0.8873 | 0.9063 | 0.2822 | 0.1419 | 1.614 | 0.1248 | 0.0913 | 1.201 |
| PEPLER-d-emb | 0.4324 | 0.1158 | 0.4678 | 0.1821 | **0.9004** | 0.9069 | 0.2675 | **0.1593** | 1.311 | 0.1262 | **0.1036** | 1.205 |

**Table 19: The performance gains/losses in existing metrics on Yelp when we use the ground-truth ratings as input (shown with "+").** The best scores of all models are underlined, and the gains/losses are marked in ↑green and ↓red, respectively.

| Method | Text Quality | | | | | | Explainability | | | | | |
| | | | | | | | | Positive | | | Negative | |
| | B1 ↑ | B4 ↑ | R1 ↑ | R2 ↑ | USR ↑ | BERT ↑ | FMR ↑ | FCR ↑ | DIV ↓ | FMR ↑ | FCR ↑ | DIV ↓ |
|---|---|---|---|---|---|---|---|---|---|---|---|---|
| PETER-d-emb | 0.3861 | 0.0716 | 0.4049 | 0.1268 | 0.7187 | 0.8882 | 0.2300 | 0.1991 | 2.232 | 0.1223 | 0.1813 | 1.372 |
| PETER-d-emb+ | _0.4083_ | _0.0938_ | _0.4261_ | _0.1622_ | _0.7534_ | _0.8914_ | _0.2410_ | _0.2249_ | _2.309_ | _0.1389_ | _0.2052_ | _1.403_ |
| | ↑5.7% | ↑31.0% | ↑5.2% | ↑27.9% | ↑4.8% | ↑0.3% | ↑4.7% | ↑12.9% | ↓3.4% | ↑13.5% | ↑13.1% | ↓2.2% |
| PEPLER-d-emb | 0.3768 | 0.0696 | 0.3934 | 0.1233 | 0.9277 | 0.8859 | 0.2199 | 0.2719 | 2.227 | 0.1081 | 0.2590 | _1.166_ |
| PEPLER-d-emb+ | _0.4006_ | _0.0884_ | _0.4175_ | _0.1552_ | _0.8269_ | _0.8904_ | _0.2225_ | _0.2599_ | _2.314_ | _0.1334_ | _0.2461_ | _1.439_ |
| | ↑6.3% | ↑27.0% | ↑6.1% | ↑25.8% | ↓10.8% | ↑0.5% | ↑1.1% | ↓4.4% | ↓3.9% | ↑23.4% | ↓4.9% | ↓23.4% |

**Table 20: The performance gains/losses in existing metrics on RateBeer when we use the ground-truth ratings as input (shown with "+").** The best scores of all models are underlined, and the gains/losses are marked in ↑green and ↓red, respectively.

| Method | Text Quality | | | | | | Explainability | | | | | |
| | | | | | | | | Positive | | | Negative | |
| | B1 ↑ | B4 ↑ | R1 ↑ | R2 ↑ | USR ↑ | BERT ↑ | FMR ↑ | FCR ↑ | DIV ↓ | FMR ↑ | FCR ↑ | DIV ↓ |
|---|---|---|---|---|---|---|---|---|---|---|---|---|
| PETER-d-emb | 0.4364 | 0.1214 | 0.4728 | 0.1885 | 0.8239 | 0.9073 | 0.2778 | 0.1294 | 1.448 | 0.1322 | 0.0818 | 1.195 |
| PETER-d-emb+ | _0.4415_ | _0.1243_ | _0.4762_ | _0.1945_ | _0.8421_ | _0.9073_ | _0.2859_ | _0.1380_ | _1.369_ | _0.1400_ | _0.0886_ | _1.130_ |
| | ↑1.1% | ↑2.3% | ↑0.7% | ↑3.1% | ↑2.2% | ↑0.0% | ↑2.9% | ↑6.6% | ↑5.4% | ↑5.9% | ↑8.3% | ↑5.4% |
| PEPLER-d-emb | 0.4324 | 0.1158 | 0.4678 | 0.1821 | 0.9004 | 0.9069 | 0.2675 | 0.1593 | 1.311 | 0.1262 | 0.1036 | 1.205 |
| PEPLER-d-emb+ | _0.4461_ | _0.1231_ | _0.4815_ | _0.1981_ | _0.9230_ | _0.9075_ | _0.2611_ | _0.1639_ | _1.287_ | _0.1325_ | _0.1098_ | _1.076_ |
| | ↑3.1% | ↑6.3% | ↑2.9% | ↑8.7% | ↑2.5% | ↑0.0% | ↓2.3% | ↑2.8% | ↑1.8% | ↑4.9% | ↑5.9% | ↑10.7% |

