# OpenReview forum: "Disentangling Likes and Dislikes in Personalized Generative Explainable Recommendation"
_ACM.org/TheWebConf/2025/Conference — WWW 2025 Poster_

### Official Review · Reviewer_Tsqz · 2024-11-20

**Novelty:** 4
**Technical Quality:** 3

**Review:**

This paper introduces a novel dataset and evaluation method designed to distinguish between users' positive and negative sentiments towards recommended items, providing new research materials for the field of explainable recommendation. The authors employ GPT-4 to automatically extract sentiment features from user reviews and generate lists of users' positive and negative features, resulting in a high-quality dataset with clear sentiment information.

In the dataset, users' ratings may not always align perfectly with the extracted sentiments. For instance, a user might give a high rating while mentioning some negative aspects. Such inconsistencies between ratings and sentiment features could potentially affect the model's learning performance. This contradictory relationship has not been explicitly addressed in the dataset, which might impact the emotional coherence of the generated model.

Moreover, and most importantly, since GPT-4 is used to generate explainable texts for creating the dataset, utilizing this dataset with GPT to generate explanations might inherently lead to optimal performance, as the dataset itself was generated by GPT. Therefore, this dataset may only be suitable for evaluating the performance of non-GPT models, significantly limiting its applicability. Additionally, when using this dataset for training, other models might imitate GPT's specific output style, leading to seemingly better performance. However, such an "advantage" would primarily stem from the structural similarity between the generated data and GPT, rather than reflecting the model's ability to produce diverse and authentic explanations.

Some minor issues:
1. The sentiment-matching score only measures whether positive/negative features are mentioned but does not evaluate whether the predicted sentiment corresponds to the described aspects or facts. In other words, it only reflects the overall sentiment tendency of the sentence. In recommendation explanations, the influence of different features may vary in importance, but Content-p/n does not account for feature weights. As a result, features with low importance might significantly affect the score, potentially misaligning with users' actual perceptions.

2. Incorporating users' predicted ratings as input can enhance the performance of the explanation text generator by providing the model with additional information. Treating rating prediction as a subtask is also effective, as it helps prevent the sentiment tendency of the generated text from deviating from the user's overall attitude. I believe the two approaches can coexist, and using ratings as input while simultaneously treating rating prediction as a subtask could yield better performance.

3. Why do models treating ratings as discrete variables generally perform better than those treating them as continuous variables? This is likely due to the nonlinear relationship between users' emotions and their ratings of items. Moreover, from the observations in Table 8, it is evident that -d-emb does not consistently outperform -c-emb.

4. Line 743: This suggests that optimal models differ depending on the nature of the dataset.
   This lacks further analysis—what specific characteristics of the dataset are related to this observation?

**Questions:**

The dataset created by the authors only distinguishes between positive and negative sentiments, without addressing the multidimensional or nuanced emotional differences that users may hold (e.g., "slightly like" vs. "strongly like"). Given the use of LLMs for extraction, could such detailed variations be captured through prompt engineering?

**Reviewer Confidence:**

3: The reviewer is confident but not certain that the evaluation is correct

**Scope:**

2: The connection to the Web is incidental, e.g., use of Web data or API

---

### Official Review · Reviewer_FfXj · 2024-11-21

**Novelty:** 5
**Technical Quality:** 6

**Review:**

This paper makes a meaningful contribution to the field of explainable recommendation systems by introducing datasets and metrics that focus on user sentiments. The paper also presents a modified version of PETER along with extensive experiments to validate their claims. However, although I can follow the claims and experiments in the paper, since I am not too familiar with explainable recommendations, I will have a confidence of 2.

## Pros
- The authors introduce datasets that explicitly disentangle users’ positive and negative opinions using an LLM, providing a more fine-grained understanding of user sentiment compared to existing datasets.
- Proposing sentiment-matching and content similarity metrics makes the evaluations more rigorous.
- The paper benchmarks state-of-the-art models and introduces variations that integrate predicted ratings as input.
- The method is well-documented, with detailed explanations of dataset construction and metrics.
- Human evaluations were conducted to validate the performance of the LLMs.
## Cons
- The proposed methods could be written using notations to be more explicit.
- For the experiments, especially in Table 11 and 12, some of the improvements seem incremental (e.g. 0.2%). Doing some significance tests could help the quantitative analysis of the models.
- Some parts of the paper are not well discussed. Please see the questions below.

**Questions:**

- On line 208, the authors mention that the model's output is restrict to 15 words. Is this achieved through prompting or just filtering out output with longer than 15 words?
- Between line 445 and 448, the authors introduce the sentiment-matching score. I am actually not quite sure about how it is computed. Are you considering all the explanations together no matter the content and measure the percentage of explanations that have the same sentiment? Using math notations to writer this out explicit would help the readers to better understand.

**Reviewer Confidence:**

2: The reviewer is willing to defend the evaluation, but it is likely that the reviewer did not understand parts of the paper

**Scope:**

4: The work is relevant to the Web and to the track, and is of broad interest to the community

---

### Official Review · Reviewer_hUQ9 · 2024-12-01

**Novelty:** 5
**Technical Quality:** 4

**Review:**

This paper addresses an important limitation in existing explainable recommender systems: the inability to align generated explanations with users’ positive and negative sentiments. The authors propose a new task focusing on sentiment alignment and introduce novel datasets specifically designed to evaluate this aspect. These datasets are constructed using a two-step process involving review summarization and aspect-based sentiment analysis, utilizing a large language model (LLM). The authors also propose new evaluation metrics to assess the alignment between generated and ground-truth explanations. Through extensive experiments, they benchmark several baseline models and demonstrate that existing metrics fail to capture sentiment alignment. They further show that providing ground-truth ratings as additional input improves the sentiment-aware performance of baseline models. The study opens new research directions for explainable recommender systems by focusing on the alignment of explanations with user sentiments.

Overall, this is a good paper, and the proposed method is novel and sound. If this paper is finally accepted, I suggest that the authors provide a more comprehensive discussion of related work on sentiment-related explainable recommendations and offer a deeper explanation of the rationale behind the dataset construction process.

1. The structure of this paper is clear and its core idea is easy to follow.

2. The existence of mixed feelings in user reviews is a well-established phenomenon. Without explicitly considering users' sentiments, explainable recommender systems may fail to align their explanations with the positive and negative features expressed in the reviews. By constructing a new dataset that explicitly includes positive and negative features, the authors provide a valuable benchmark for evaluating sentiment-aware explanations, addressing this critical gap effectively.

3. The paper successfully demonstrates through experiments that existing evaluation metrics are insufficient for assessing sentiment-aware explanations. Furthermore, the authors introduce new evaluation metrics specifically designed to address this limitation, providing a more accurate and comprehensive framework for evaluating sentiment alignment in explainable recommendations.

4. The case study effectively highlights the significance of incorporating users’ sentiments into the evaluation process to accurately assess the quality of explanations.


Weaknesses.
1. A potential weakness is the decision to perform review summarization as a mandatory first step in dataset construction. This approach might risk omitting certain positive or negative features, potentially compromising the accuracy of subsequent feature extraction. It is unclear whether the authors have considered this issue or explored alternative approaches, such as multi-round extraction, to mitigate these limitations.

2. Requiring the extracted features to strictly match the exact words or phrases in the output of the review summarization task might result in the loss of implicit features that are not explicitly mentioned in the original review text but can be inferred from the context.

3. The authors may have overlooked some relevant related work on sentiment-aware explainable recommender systems, which also take users' sentiments into account, such as Ref.
[1] Park S J, Chae D K, Bae H K, et al. Reinforcement learning over sentiment-augmented knowledge graphs towards accurate and explainable recommendation[C]//Proceedings of the fifteenth ACM international conference on web search and data mining. 2022: 784-793.
[2] Xie F, Wang Y, Xu K, et al. A Review-Level Sentiment Information Enhanced Multitask Learning Approach for Explainable Recommendation[J]. IEEE Transactions on Computational Social Systems, 2024.

4. In the paper, references [21] and [22] refer to the same paper. The correct reference for PETER [21] should be as follows.
[3] Lei Li, Yongfeng Zhang, and Li Chen. 2021. Personalized transformer for explainable recommendation. In Proceedings of the 59th Annual Meeting of the Association for Computational Linguistics. 4947–4957.

**Questions:**

1. This approach might risk omitting certain positive or negative features, potentially compromising the accuracy of subsequent feature extraction. It is unclear whether the authors have considered this issue or explored alternative approaches, such as multi-round extraction, to mitigate these limitations.

2. Requiring the extracted features to strictly match the exact words or phrases in the output of the review summarization task might result in the loss of implicit features that are not explicitly mentioned in the original review text but can be inferred from the context.

3. Is it ensured that all positive and negative features have been extracted from the original review text?

4. Are the extracted positive and negative features non-redundant? For instance, has independence testing been conducted to verify their uniqueness?

**Reviewer Confidence:**

3: The reviewer is confident but not certain that the evaluation is correct

**Scope:**

4: The work is relevant to the Web and to the track, and is of broad interest to the community

---

### Official Review · Reviewer_rhoR · 2024-12-02

**Novelty:** 5
**Technical Quality:** 5

**Review:**

This paper highlights that current text-based evaluation methods for explainable recommendations are hard to capture users' sentimental preferences, such as liking or disliking specific aspects of items. To address this, the authors propose a new evaluation method that leverages Large Language Models (LLMs) to generate review summaries and identify personalized positive and negative opinions about items. The authors also plan to release a new dataset if the paper is accepted. Experiments show that their method outperforms commonly used metrics like BLEU, ROUGE, and BERTScore in distinguishing explainable recommendation models.

Pros:
- The authors promise to release the code and datasets upon acceptance and include the prompts in the Appendix, enhancing reproducibility.
- The paper involves a relevant and important topic: evaluating recommendation explanations with a focus on user preferences, which is likely to attract significant interest in the research community.
- Human annotators are invited in assessing the outputs generated by LLMs, ensuring the reliability of the experimental results.
- The authors conducted multiple experiments to demonstrate the effectiveness of each component in their proposed evaluation method.

Cons:
- The paper could benefit from addressing some specific questions and suggestions, as detailed in the Questions Section.

**Questions:**

- It would be beneficial to include human annotators in evaluating the "Informative" metric [1] in addition to Factual and Context-p/n. For example, in the case study, a statement like "user loves the delicious food" is too general for a generated explanation. The "Informative" metric ensures that the explanation is specific to the user-item pair. Ideally, the explanation should cover all relevant positive and negative features without being overly verbose.
- Since LLMs can produce different outputs for the same input, the authors should report the average results over multiple trials.
- Table 6 shows that GPT-4 and human annotators generally make similar judgments. The authors should confirm that the annotators did not rely on any LLMs for evaluation. Additionally, it may be necessary to explore alternative methods to assess annotator quality. Furthermore, as each review is evaluated by only one annotator, cross-validation between annotators is not possible.
- In the experiment setup, the authors use RoBERTa-large to calculate the Content-p/n metrics in a way similar to BERTScore. However, the performance table only reports BERTScore results, not RoBERTa-large. The authors should explain this omission. Moreover, they should clarify why this paper cites the P5 model but does not include it as a reference explainable recommendation model.

[1] Explainable and coherent complement recommendation based on large language models.

**Reviewer Confidence:**

3: The reviewer is confident but not certain that the evaluation is correct

**Scope:**

4: The work is relevant to the Web and to the track, and is of broad interest to the community

---

### Official Review · Reviewer_UUMe · 2024-12-02

**Novelty:** 2
**Technical Quality:** 3

**Review:**

This work proposed a new benchmark (set of datasets) on the prediction of post purchase sentiments, which is kinda similar to the CTR task but with sentiments rather than simple score or regression. Having such benchmarks is definitely beneficial for the research community, and can inspire many new research under this topic. Nonetheless, I do have a couple concerns regarding this work:

1. The datasets are quite tiny, with at most 20K entities and half million interactions.
2. The ground-truth explanations are just simply outputs of LLMs via prompting.

**Questions:**

n/a

**Reviewer Confidence:**

2: The reviewer is willing to defend the evaluation, but it is likely that the reviewer did not understand parts of the paper

**Scope:**

3: The work is somewhat relevant to the Web and to the track, and is of narrow interest to a sub-community